# A Single-Chain Mpox mRNA Vaccine Elicits Protective Immune Response in Mice

**DOI:** 10.3390/vaccines13050514

**Published:** 2025-05-13

**Authors:** Qian Xu, Rong-Rong Zhang, Mei Wu, Jie Zhang, Zu-Xin Wang, Hang Chi, Chao Zhou, Xiao-Chuan Xiong, Hai-Tao Liu, Cheng-Feng Qin, Qing Ye

**Affiliations:** 1School of Basic Medical Sciences, Wenzhou Medical University, Wenzhou 325035, China; xq15703029959@163.com (Q.X.); 18708849270@163.com (J.Z.); 2State Key Laboratory of Pathogen and Biosecurity, Academy of Military Medical Sciences, Beijing 100071, China; 15256241239@163.com (R.-R.Z.); wu_mei98@163.com (M.W.); 15620785009@163.com (Z.-X.W.); ch_amms@163.com (H.C.); zhouchaozhixia@163.com (C.Z.); xxc1553121917@163.com (X.-C.X.); haitao816@foxmail.com (H.-T.L.)

**Keywords:** mpox, mRNA vaccine, single chain

## Abstract

**Background:** The re-emerging mpox virus (MPXV) has spread to numerous countries and raised global concern. There is an urgent need for a safe and effective mRNA vaccine candidate against MPXV infection. Previously, we developed a penta-component mRNA vaccine that contained five distinct antigen-encoded mRNAs encapsulated within lipid nanoparticles (LNPs). Here, we sought to develop a single-chain mRNA vaccine that encodes antigens derived from both intracellular mature virion (IMV) and extracellular enveloped virion (EEV). **Methods:** A single-chain mRNA vaccine encoding a fusion protein comprising the ectodomains of M1R (eM1R) and A35R (eA35R) (MPXV^eM1-eA35^) was developed and characterized, while an admixed formulation of two individual mRNA-LNPs encoding separate antigens was developed as the control (MPXV^eM1+eA35^). Meanwhile, based on the same strategy, we designed a single-chain mRNA vaccine encoding dimeric antigens (MPXV^eM1-eA35-Fc^). Mice were immunized with two doses of the candidate vaccines, and both humoral and cellular immune responses were evaluated. The protective efficacy of the candidate vaccines was evaluated based on body weight monitoring and tissue viral load measurement after challenge with vaccinia virus (VACV). **Results:** Immunization with two doses of MPXV^eM1-eA35^ elicited robust levels of neutralizing antibodies and antigen-specific cellular immune response. Importantly, MPXV^eM1-eA35^ demonstrated protective efficacy in a VACV challenge mouse model and showed superior capacity in preventing weight loss post-challenge compared to MPXV^eM1+eA35^. Similarly, MPXV^eM1-eA35-Fc^ exhibited comparable or superior immunogenicity and protective efficacy compared to the admixed formulations. **Conclusions:** The single-chain mRNA vaccine elicited a protective immune response in mice, offering significant advantages in terms of manufacturing processes and quality control. Our single-chain mRNA vaccine platform presents a promising strategy for the next generation design of mpox vaccines and contributes to the mitigation of MPXV endemic worldwide.

## 1. Introduction

Mpox is a zoonotic viral disease, with typical symptoms including rash, fever, fatigue, muscle pain, and lymphadenopathy [1]. The mpox virus (MPXV), which causes mpox, is a double-stranded DNA virus belonging to the Orthopoxvirus genus of the Poxviridae family [2]. MPXV was initially endemic in the tropical rainforest regions of West and Central Africa [3,4,5]. Since 2022, MPXV infection cases have been reported in multiple non-endemic countries. The World Health Organization (WHO) has declared mpox a public health emergency of international concern (PHEIC) twice. To date, MPXV is actively circulating and evolving in some regions [6], indicating that the threat of a continued mpox epidemic remains. Therefore, developing effective prevention and control strategies is crucial. 

Vaccination is one of the most effective strategies to combat MPXV infection. To date, only two smallpox vaccines have been authorized by the U.S. Food and Drug Administration (FDA) for preventing MPXV infection: the live vaccinia virus vaccine ACAM 2000 and the attenuated non-replicating smallpox vaccine JYNNEOS [7]. JYNNEOS is approved for preventing both smallpox virus and MPXV infections in high-risk adults aged 18 years and older. It is currently prioritized for healthcare workers and individuals with confirmed or suspected exposure to mpox. Recently, reduced levels of neutralizing antibodies against MPXV were observed in healthy individuals after vaccination [8,9]. The ACAM 2000 vaccine is authorized for high-risk populations aged 2 years and older with normal immune function. However, due to its association with myocarditis and pericarditis, it is contraindicated for immunocompromised individuals, pregnant women, or those who are breastfeeding [10,11]. Thus, it is highly necessary to develop novel vaccines specifically targeting MPXV infection.

mRNA vaccines have demonstrated significant potential in combating viral infectious diseases, including mpox [12,13]. Several successfully developed COVID-19 mRNA vaccines employ LNP delivery technology, demonstrating the efficacy of LNPs in stabilizing mRNA and effectively delivering it into cells [12,14,15]. The infection process of MPXV involves two infectious viral particles: the intracellular mature virus (IMV) and the extracellular enveloped virus (EEV) [16,17]. The IMV surface protein L1R and EEV surface protein A33R from VACV have been demonstrated to confer protection against poxvirus infection in mice [18]. VACV L1R (homologous to MPXV M1R) has been shown to induce potent neutralizing responses and protect mice from lethal VACV infection [19]. VACV A33R (homologous to MPXV A35R) plays a critical role in viral cell-to-cell transmission [20] and has been shown to enhance the protective efficacy of L1R in mice [21]. Additional studies have revealed that monoclonal antibodies targeting M1R and A35R of MPXV protect mice from poxvirus challenge [8]. Multiple antigen combinations targeting both IMV and EEV have demonstrated superior protective efficacy compared to a single antigen against poxvirus infection [16,22]. Recently, several studies on mpox mRNA vaccines have explored the use of mRNA mixtures expressing individual antigens [23,24,25,26,27,28]. However, these approaches are often associated with high costs and stringent quality control requirements. 

In this study, we developed a single-chain mRNA vaccine encoding the ectodomains of M1R and A35R of MPXV. The vaccine candidate elicited high levels of neutralizing antibodies and antigen-specific cellular immune response. Based on a VACV challenge mouse model, the single-chain mRNA vaccine showed a non-inferior protective efficacy compared to the admixed formulation of individual mRNA-LNPs. Notably, the single-chain mRNA vaccine offers significant advantages in terms of manufacturing processes and quality control standardization, representing a promising strategy for next-generation mpox vaccine development.

## 2. Materials and Methods

### 2.1. Cells

The cell lines used in this work, including HEK293T and BS-C-1, were acquired from the American Type Culture Collection (ATCC). HEK293T cells were maintained at 37 °C with 5% CO_2_, using DMEM culture medium containing 10% fetal bovine serum (FBS) and 1% HEPES buffer. For BS-C-1 cell culture, Minimum Essential Medium (MEM, Procell, Wuhan, China) supplemented with 10% FBS and 1% penicillin–streptomycin was employed under identical incubation conditions.

### 2.2. Virus

The vaccinia virus Western Reserve strain (VACV-WR, GenBank: AY243312.1) was provided by the Institute of Microbiology, Chinese Academy of Sciences. Viral propagation was performed in BS-C-1 cell monolayers. At 48 h post-infection, infected cells were harvested in phosphate-buffered saline (PBS, Solarbio, Beijing, China) and subjected to three freeze–thaw cycles for cell lysis. Viral titers were determined through plaque assays using BS-C-1 cells. Prior to use, fetal bovine serum (FBS, Gibco, Grand Island, NE, USA) was heat-inactivated at 56 °C for 30 min.

### 2.3. mRNA Preparation

DNA sequences of MPXV A35R, M1R were optimized and synthesized by Sangon Biotech (Shanghai, China) and inserted into the plasmid ABOP-028 (GENEWIZ, Nanjing, China). The mRNA was produced through in vitro transcription mediated by T7 RNA polymerase using linearized DNA as the template.

### 2.4. Identification of mRNAs In Vitro Expression

In vitro mRNA expression was detected in HEK293T cells using an immunofluorescence assay (IFA) and Western blot. mRNA (1 μg) was transfected into monolayer HEK293T cells using Lipofectamine™ MessengerMAX™ (Thermo Fisher Scientific, Waltham, MA, USA). For Western blot analysis, at 24 h post-transfection, cell lysates were subjected to reducing and non-reducing treatments and then separated by polyacrylamide gel (Sangon Biotech, Shanghai, China) using different voltages, followed by protein transfer to PVDF membranes at a constant current. The membranes were incubated with 5% non-fat milk for 1 h and then probed with anti-Flag antibody at 37 °C for 1 h. Subsequently, the membranes were incubated with goat anti-mouse IgG-HRP antibody (1:10,000, ZSGB-Bio, Beijing, China) at room temperature for 1 h. For IFA, cells were fixed with acetone–methanol (3:7) at room temperature for 30 min and blocked with BSA. The cells were incubated with anti-Flag antibody at 37 °C for 2 h, followed by Alexa Fluor 488-conjugated anti-mouse antibody (1:200, Thermo Fisher Scientific, Waltham, MA, USA) at 37 °C for 1 h. Images were captured using a fluorescence microscope.

### 2.5. LNP Encapsulation of mRNA

LNP formulations were prepared as described previously [12]. The encapsulation process was performed using a NanoAssemblr^®^ Ignite+ microfluidic mixer (Precision Nanosystems, Vancouver, BC, Canada), wherein mRNA was complexed with a lipid mixture comprising ionizable lipid, DSPC, cholesterol, and PEG–lipid (molar ratio 50:10:38.5:1.5; Abogenbio, Suzhou, China). Subsequent buffer exchange was achieved through tangential flow filtration (20 kDa RC membrane; Thermo Scientific, Waltham, MA, USA) against 10 volumes of PBS (pH 7.4), followed by sterile filtration (0.22 μm). Quality control assessments included particle size distribution analysis and RNA quantification, with final products stored at 2–8 °C.

### 2.6. Vaccination, VACV Challenge, and Sample Collection Protocols

Six-week-old female BALB/c mice were obtained from Vital River Animal Technology Co., Ltd. (Beijing, China), and randomly divided into groups. Mice were immunized via intramuscular (IM) injection with MPXV^eM1^ (n = 15), MPXV^eA35^ (n = 15), MPXV^eM1eA35^ (n = 15), MPXV^eM1+eA35^ (n = 15), MPXV^eM1-eA35-Fc^ (n = 15), and MPXV^eM1+eA35-Fc^ (n = 15) (total dose of 5 μg per vaccine), followed by a booster immunization with the same dose and method three weeks later. The mice were injected with empty LNPs (without mRNA) as the placebo group (n = 15). Serum samples were collected on the 35th day after the initial immunization to detect MPXV-specific IgG antibodies and neutralizing antibody responses (n = 5). Spleens were harvested on the 35th day after the initial immunization to assess cellular immune responses via ELISpot (n = 5). On the 42nd day after the initial immunization, the immunized mice were challenged intranasally with VACV-WR (1 × 10⁵ PFU) (n = 10), and body weight changes were monitored for 14 days (n = 5). Tissues were collected on day 7 post-challenge for the quantification of viral DNA levels (n = 5). All animal procedures were reviewed and approved by the Animal Experiment Committee of the Laboratory Animal Center, Academy of Military Medical Sciences (AMMS) (approval number: IACUC-DWZX-2022-055).

### 2.7. Enzyme-Linked Immunosorbent Assay (ELISA)

Prior to analysis, serum samples were heat-inactivated at 56 °C for 30 min. ELISA plates were coated overnight at 4 °C with 50 ng/well of A35R or M1R antigen in 10× coating buffer (Solarbio, Beijing, China), followed by blocking with 5% non-fat milk at 37 °C for 2 h. Serially diluted serum samples were then incubated in the plates for 2 h at 37 °C, after which the plates were washed and probed with HRP-conjugated goat anti-mouse IgG secondary antibody (1:10,000, ZSGB-Bio) for 1 h at 37 °C. Following additional washes, color development was initiated by adding TMB substrate (Cwbio, Beijng, China) and stopped after approximately 15 min using stop solution (Solarbio). Absorbance at 450 nm was measured using a Synergy H1 hybrid multimode microplate reader (BioTek, Winooski, VT, USA), with antibody titers determined as the reciprocal of the highest serum dilution yielding absorbance values that exceed twice the background level.

### 2.8. Plaque Reduction Neutralization Test (PRNT)

Serum samples were heat-inactivated (56 °C, 30 min) prior to three-fold serial dilution in RPMI 1640 medium supplemented with 2.5% FBS. The diluted sera were combined with an equal volume of VACV suspension (300 PFU/mL) and incubated (37 °C, 90 min) to facilitate antibody-mediated neutralization. The serum–virus complexes were then adsorbed to BS-C-1 cell monolayers (37 °C, 90 min). Post-adsorption, the cells were maintained under semisolid overlay medium (0.5% methylcellulose in DMEM with 2.5% heat-inactivated FBS) for 48 h at 37 °C. Following fixation with 4% paraformaldehyde and 1% crystal violet staining, the viral plaques were enumerated. The PRNT_50_ titers were determined through Spearman–Karber analysis.

### 2.9. Enzyme-Linked ImmunoSPOT (ELISpot)

The cellular immune responses in vaccinated mice were assessed using an ELISpot assay. Following pre-coating with IFN-γ capture antibody, the plates were conditioned with RPMI 1640 medium containing 10% FBS for 30 min at room temperature. Freshly isolated mouse splenocytes were stimulated with M1R and A35R peptide pools in the prepared plates and cultured for 36 h at 37 °C under a 5% CO_2_ atmosphere, with appropriate negative and positive controls established in parallel. Subsequent to PBS washing, detection was performed by sequential incubation with biotinylated anti-mouse IFN-γ antibody (2 h, room temperature) and streptavidin–ALP conjugate (1 h, room temperature). Following substrate development, spot-forming cells were quantified using an automated ELISpot reader after image acquisition.

### 2.10. Real-Time Quantitative PCR (qPCR) Assay

Viral nucleic acids were extracted by using the DNeasy Blood & Tissue Kit (QIAGEN, Hilden, Germany). PCR was conducted using a Probe qPCR Mix containing VACV-F (5′-GGCAATGGATTCAGGGATATAC-3′), VACV-R (5′-ATTTATGAATAATCCGCCAGTTAC-3′), and VACV-P (5′-CAATGTGTCCGCTGTTTCCGTTAATAAT-3′) [23] in a LightCycler^®^ 480 Instrument (Roche Diagnostics Ltd., Basel, Switzerland). The standard curve was produced by ten-fold serial dilutions of VACV DNA. Thus, the viral DNA of biological samples was quantified from the standard curve.

### 2.11. Statistical Analysis

Statistical analyses were carried out using GraphPad Prism version 10.0.0 (GraphPad software). Data are presented as mean ± SEM.

## 3. Results

### 3.1. Design and Characterization of a Single-Chain Mpox mRNA Vaccine Encoding the Ectodomains of M1R and A35R

To rationally design a single-chain mpox mRNA vaccine, an mRNA construct encoding a fusion protein comprising the ectodomains of M1R (eM1R) and A35R (eA35R) was developed, named eM1-eA35. Meanwhile, mRNA molecules encoding either a single eM1R or eA35R, respectively, were also constructed (Figure 1a). Equal amounts of antigen-encoding mRNAs were transfected into HEK293T cells. Protein expression was verified by IFA at 24 h post-transfection using an anti-Flag antibody, with successfully transfected cells exhibiting distinct green fluorescence signals (Figure 1b). Subsequently, the antigen-encoded mRNAs were encapsulated into LNPs to formulate the final vaccine candidates. Specifically, MPXV^eM1-eA35^ was a single-chain mRNA vaccine containing LNP-encapsulated eM1-eA35, while MPXV^eM1+eA35^ consisted of a mixture of MPXV^eM1^ and MPXV^eA35^ formulations and was prepared as the control (Figure 1a). Dynamic light-scattering analyses showed uniform particle sizes of mRNA-LNPs ranging from 109 to 118 nm with a polydispersity index (PDI) below 0.1 (Appendix A).

### 3.2. Single-Chain mRNA Vaccine Candidate Effectively Activates Humoral Immune Response and T Cell Immune Response

To evaluate the immunogenicity of eM1R and eA35R antigens, groups of six-week-old BALB/c mice were intramuscularly immunized with MPXV^eM1^ or MPXV^eA35^ on days 0 and 21, respectively. Empty LNPs were used as the placebo. Serum samples were collected on day 35 post-initial immunization to assess antigen-specific IgG antibody and neutralizing antibody titers against VACV (Appendix A). The results showed that MPXV^eM1^ elicited a high antibody titer against M1R, whereas MPXV^eA35^ failed to trigger the production of IgG antibodies against A35R (Appendix A). The neutralization capacity of the immunized sera was assessed using the PRNT. MPXV^eM1^ elicited a strong neutralizing antibody response against VACV, whereas MPXV^eA35^ failed to induce neutralizing antibodies against VACV (Appendix A). In addition, cellular immune responses were evaluated using the ELISpot assay. The results revealed that splenocytes derived from the immunized mice exhibited a significant induction of IFN-γ upon stimulation with the antigen peptide pools (Appendix A). 

Next, mice were immunized with MPXV^eM1-eA35^ and MPXV^eM1+eA35^ using the same experimental protocol as described above (Figure 1c). The results showed that the single-chain MPXV^eM1-eA35^ effectively elicited IgG antibodies against M1R and A35R, whereas MPXV^eM1+eA35^ only induced detectable antibodies against M1R (Figure 1d,e). Moreover, high levels of neutralizing antibodies against VACV were detected in animals immunized with both MPXV^eM1-eA35^ and MPXV^eM1+eA35^ (Figure 1f). Furthermore, cellular immune responses were evaluated using the ELISpot assay. Splenocytes from mice vaccinated with MPXV^eM1-eA35^ or MPXV^eM1+eA35^ were stimulated with the M1R and A35R antigen peptide pools, resulting in significant induction of IFN-γ. The results demonstrated that both MPXV^eM1-eA35^ and MPXV^eM1+eA35^ elicited antigen-specific cellular immune responses compared to the placebo group (Figure 1g,h).

### 3.3. Single-Chain mRNA Vaccine Protects Mice from VACV Challenge

To evaluate the protective efficacy of the single-chain mRNA vaccine candidate, vaccinated mice were intranasally challenged with VACV-WR at a dose of 10^5^ PFU on day 42 after initial immunization. Body weight changes were recorded for 14 days post-inoculation (dpi). The placebo-immunized mice showed a 19% reduction in body weight at 7 dpi. Mice immunized with MPXV^eM1-eA35^ showed only minimal body weight fluctuations (<5%). MPXV^eM1+eA35^-immunized mice exhibited a transient 7.1% weight loss at 5 dpi (Figure 2a). Additionally, viral load was detected at 7 dpi. The viral DNA levels in the throat swab, lung, and nasal mucosa of all vaccine groups were significantly lower than those in the placebo group. Notably, the viral DNA was completely eradicated in the tissues of the MPXV^eM1-eA35^-immunized group, while a marginal amount of viral DNA was detected in the MPXV^eM1+eA35^-immunized group (Figure 2b–d). These results indicate that both MPXV^eM1-eA35^ and MPXV^eM1+eA35^ can protect mice from VACV-WR challenge, and the single-chain MPXV^eM1-eA35^ enables mice to exhibit reduced weight loss following VACV challenge.

### 3.4. Design and Characterization of a Single-Chain Mpox mRNA Vaccine Encoding the Dimeric Antigens

To further design a single-chain mRNA vaccine encoding dimeric antigens, the human IgG1 Fc was inserted downstream of the antigen coding sequence to facilitate the Fc-mediated dimerization of the encoded proteins. The constructed mRNA molecules were designated as eM1-eA35-Fc, eM1R-Fc, and eA35R-Fc (Figure 3a). Capped mRNA was transfected into HEK293T cells, and the in vitro expression of each mRNA was analyzed by Western blot and IFA (Figure 3b,c and Appendix A). The results demonstrated that the expressed proteins successfully formed dimeric configurations (Figure 3c). The eM1-eA35-Fc encoding the dimeric eM1R and eA35R was further encapsulated into LNPs to formulate the single-chain mRNA vaccine MPXV^eM1-eA35-Fc^, while MPXV^eM1+eA35-Fc^ consisted of a mixture of MPXV^eM1-Fc^ and MPXV^eA35-Fc^ formulations and was prepared as the control. Dynamic light scattering analysis demonstrates that both MPXV^eM1-eA35-Fc^ and MPXV^eM1+eA35-Fc^ exhibit uniform particle distributions, with particle sizes ranging from 109 to 116 nm (Appendix A).

### 3.5. MPXV^eM1-eA35-Fc^ Effectively Activates Humoral Immune Response and T Cell Immune Response

To evaluate the immunogenicity of MPXV^eM1-eA35-Fc^ and MPXV^eM1+eA35-Fc^, six-week-old BALB/c mice were immunized at days 0 and 21. Serum samples were collected on day 35 post-initial immunization (Figure 3d), and ELISA results demonstrated that both MPXV^eM1-eA35-Fc^ and MPXV^eM1+eA35-Fc^ elicited specific antibodies against M1R and A35R (Figure 3e,f). Notably, both MPXV^eM1-eA35-Fc^ and MPXV^eM1+eA35-Fc^ induced high levels of anti-M1R antibody responses (Figure 3f). MPXV^eM1-eA35-Fc^ elicited IgG antibodies against A35R in all mice, whereas MPXV^eM1+eA35-Fc^ induced detectable anti-A35R antibodies in only a portion of the mice. (Figure 3e). We further utilized the serum samples to assess neutralizing antibodies against VACV. The results showed that both MPXV^eM1-eA35-Fc^ and MPXV^eM1+eA35-Fc^ elicited potent neutralizing antibody responses against VACV (Figure 3g).

In addition, splenocytes were isolated from mice to evaluate the cellular immune responses induced by vaccination. The results showed that both the MPXV^eM1-eA35-Fc^ and MPXV^eM1+eA35-Fc^ groups elicited strong Th1-biased cellular immune responses upon stimulation with A35R and M1R peptide pools (Figure 3h,i).

### 3.6. MPXV^eM1-eA35-Fc^ Encoding Dimeric Antigens Protects Mice from VACV Challenge

Finally, we assessed the protective efficacy of the vaccines by challenging the vaccinated mice with VACV-WR. Through monitoring the body weight of mice post-viral challenge, the vaccine groups exhibited less weight loss compared to the placebo group. Notably, the MPXV^eM1-eA35-Fc^ group showed only a slight weight reduction of less than 5%, from which they quickly recovered, whereas the MPXV^eM1+eA35-Fc^ group experienced a 10% decrease in body weight at 7 dpi (Figure 4a). Subsequently, the viral load in the tissues of mice was measured on day 7 after viral challenge. Both the MPXV^eM1-eA35-Fc^ and MPXV^eM1+eA35-Fc^ immunization groups demonstrated excellent viral clearance capabilities (Figure 4b–d). The results indicated that both MPXV^eM1-eA35-Fc^ and MPXV^eM1+eA35-Fc^ provided effective protection against VACV-WR challenge in mice, while MPXV^eM1-eA35-Fc^ resulted in less weight loss in mice following VACV attack.

## 4. Discussion

Previous studies have explored multi-antigen vaccines in the form of mixtures [23,24,25,26,27,29], in which multiple mRNAs encoding individual antigens are combined. However, this vaccine format comes with high manufacturing costs, complex processes, and challenging quality control, which reduce vaccine accessibility. Therefore, our research aimed to develop a single-chain mRNA vaccine encoding the extracellular domains of both A35R and M1R antigens. This approach simplifies vaccine design, thereby reducing manufacturing costs and improving vaccine accessibility.

In this study, we first designed two types of vaccine encoding the ectodomains of M1R and A35R. One is a single-chain mRNA vaccine encoding both antigens (MPXV and MPXV^eM1-eA35^), and the other is a mix of two separate mRNA-LNP formulations, each encoding a distinct antigen (MPXV^eM1+eA35^). The results showed that both of the candidate vaccines successfully induced effective neutralizing antibodies and cellular immune responses. Notably, when challenged with VACV, MPXV^eM1-eA35^ demonstrated stronger viral clearance capability compared to MPXV^eM1+eA35^, along with a superior ability to protect mice from weight loss. Additionally, we introduced the human IgG1 Fc structure into the monomeric vaccines to construct dimeric vaccines (MPXV^eM1-eA35-Fc^ and MPXV^eM1+eA35-Fc^). The results showed that both dimeric vaccines exhibited excellent immunogenicity and protective efficacy in mice. Notably, MPXV^eM1-eA35-Fc^ exhibited only a mild decrease in body weight with rapid recovery, whereas MPXV^eM1+eA35-Fc^ displayed more pronounced changes in body weight after VACV challenge. These results support the efficacy of the single-chain mRNA vaccine candidates, demonstrating non-inferior protective efficacy compared to the admixed formulation of individual mRNA-LNPs.

Previous studies found that immunization with the soluble region of A35R (sA35R) alone did not produce A35R-specific antibodies [30], and this study validated this observation. However, regardless of whether they were in monomeric or dimeric forms, single-chain vaccines (MPXV^eM1-eA35^ and MPXV^eM1-eA35-Fc^) showed high levels of A35R antibodies, while MPXV^eM1+eA35^ still failed to induce A35R-specific antibodies. The absence of antigen-specific antibodies may result from the loss of native conformation or suboptimal molecular size for effective immune recognition, which may be partially overcome when eA35R is fused with eM1R. The mechanisms through which different A35R antigen forms modulate immune responses deserve further investigation. Moreover, CD8^+^T cells are a critical component of immunity against many viral infections. Studies have shown that the expression of IFN-γ by CD8^+^T cells limits virus-induced lung pathology and dissemination to visceral tissues [31]. It has been reported that Th1-biased antigen-specific cellular immunity was detected in mice following immunization with the mpox mRNA vaccine [23]. Similarly, our vaccine also demonstrated significant IFN-γ responses targeting A35R and M1R.

In subsequent studies, it is necessary to evaluate the protective efficacy of the candidate mRNA vaccines in other animal models, particularly in MPXV infection animal models and non-human primate models, and to conduct longitudinal monitoring of antibody persistence to characterize their long-term protective potential. Additionally, assessing the immunoprotective efficacy of single-chain mRNA vaccines encoding multiple MPXV antigens is worth attention in order to validate the effectiveness of the single-chain mRNA vaccine platform.

## 5. Conclusions

In summary, this study systematically evaluated the immunogenicity and protective efficacy of MPXV vaccine candidates encoding monomeric antigens (single-chain vs. mixture of mRNAs) or dimeric antigens (single-chain vs. mixture of mRNAs), successfully developing a highly effective, low-cost, and easily quality-controlled single-chain mRNA candidate vaccine. This vaccine represents a promising and effective strategy for mpox prevention and control.

## Figures and Tables

**Figure 1 vaccines-13-00514-f001:**
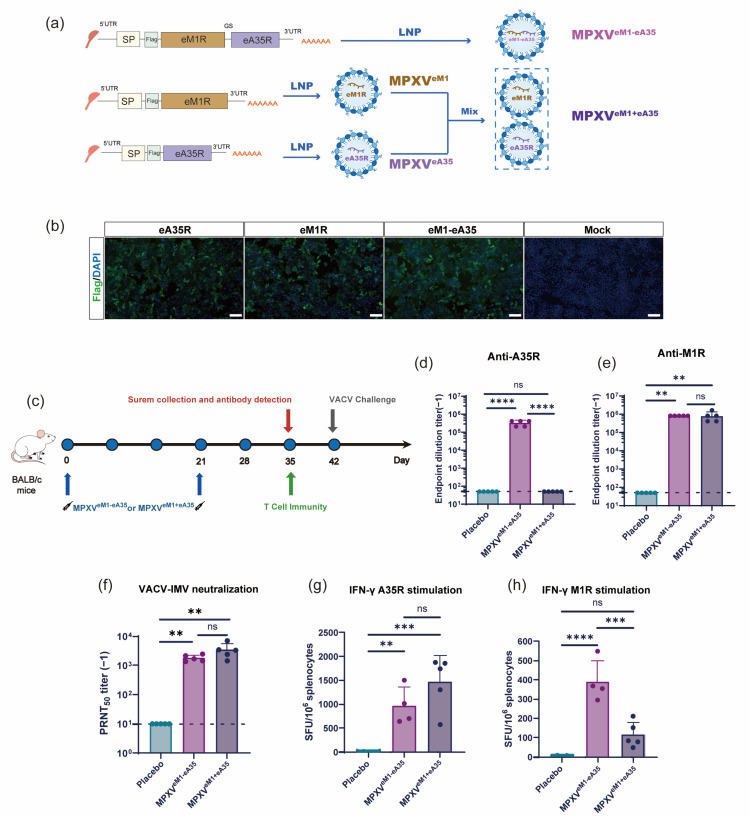
**Design and immunogenicity evaluation of mRNA vaccine candidates:** (**a**) Design and preparation scheme of mRNA vaccines: eA35R and eM1R correspond to the extracellular domains of MPXV proteins A35R and M1R, respectively. SP represents the signal peptide of IL-6. The protein domains were connected by a flexible Glycine–Serine linker. (**b**) Detection of target protein expression via indirect IFA. Different mRNAs were transfected into 293T cells. Protein expressions were confirmed by IFA using a Flag antibody. Protein-expressing cells display green fluorescence. Scale bar: 100 μm. (**c**) Immunization schedule for mRNA vaccine candidates. Female BALB/c mice were intramuscularly injected with mRNA vaccines or a placebo, followed by a booster with the same dose three weeks later. Serum and splenocyte samples were collected at specified time points. (**d**,**e**) ELISA measurement of A35R- and M1R-specific IgG antibody titers. (**f**) PRNT measurement of neutralizing antibody levels against VACV in mouse serum. (**g**,**h**) ELISpot assessment of cellular immune responses induced by mRNA vaccine candidates. Splenocytes from BALB/c mice were harvested on day 14 after the final immunization and stimulated with peptide pools of A35R (**g**) and M1R (**h**) to measure the secretion of cytokines IFN-γ (**g**,**h**). Data are presented as mean ± SEM. Statistical significance was analyzed using the Mann–Whitney test or one-way ANOVA with multiple-comparison tests (ns, not significant, ** *p* < 0.01, *** *p* < 0.001, **** *p* < 0.0001).

**Figure 2 vaccines-13-00514-f002:**
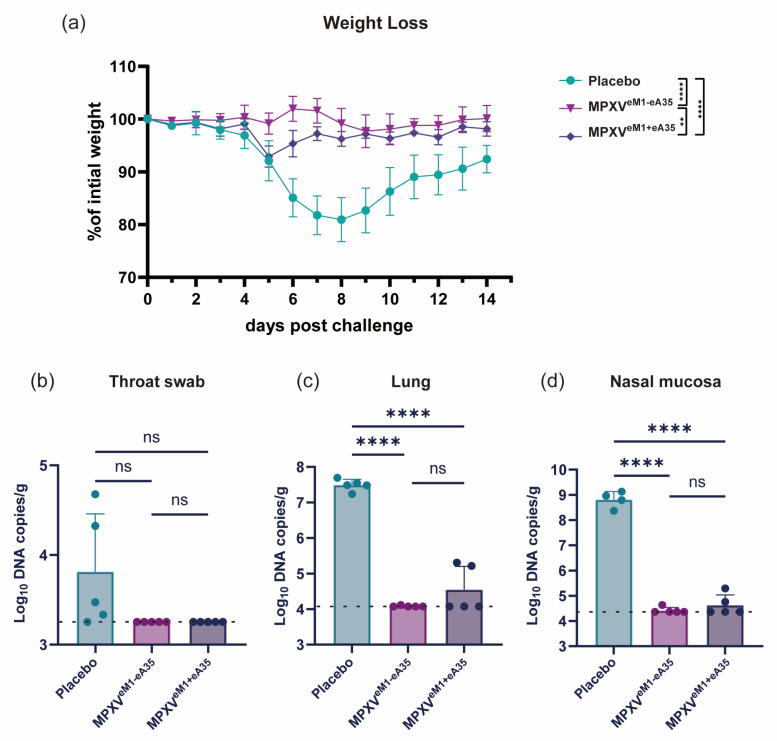
**mRNA vaccine candidates protect mice against VACV challenge:** (**a**) Mice immunized with the mRNA vaccine candidates or a placebo were intranasally inoculated with 10^5^ PFU of VACV, and changes in body weight were monitored over 14 dpi. (**b**–**d**) On day 7 after viral challenge, the mice were euthanized. Viral genome copies in throat swabs (**b**), lungs (**c**), and nasal mucosa (**d**) were detected by qPCR. Data are presented as mean ± SEM. Statistical significance was evaluated using two-way ANOVA with multiple-comparison test or one-way ANOVA. (ns, not significant, ** *p* < 0.01, and **** *p* < 0.0001).

**Figure 3 vaccines-13-00514-f003:**
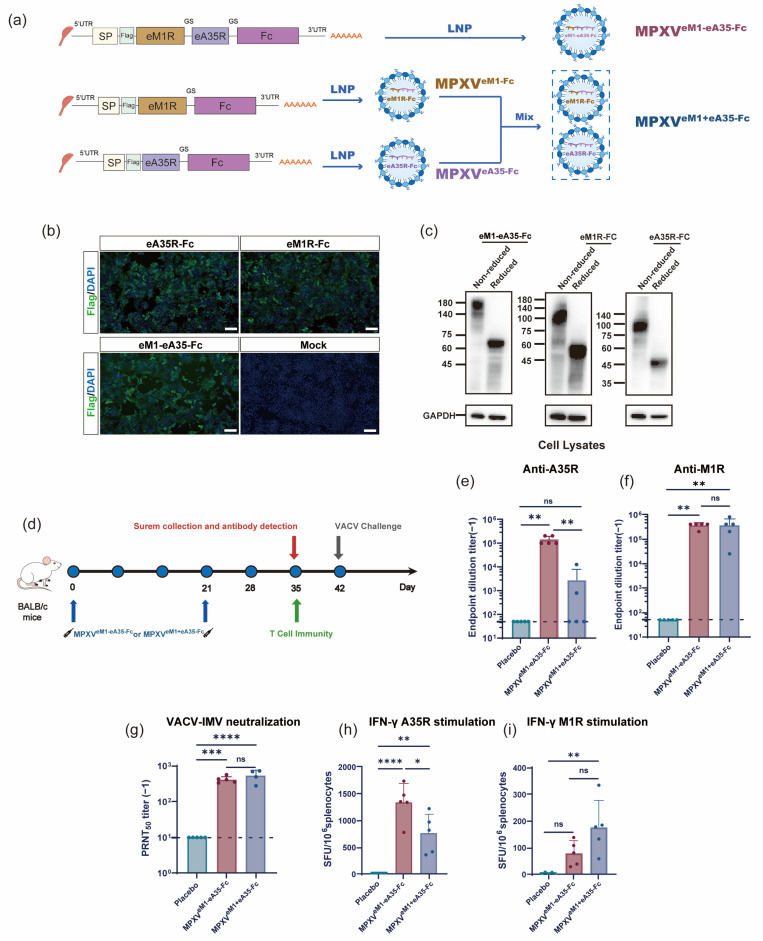
**Design and immunogenicity evaluation of mRNA vaccine candidates encoding dimeric antigens:** (**a**) Design and preparation scheme of mRNA vaccine candidates encoding dimerized antigens. eA35R and eM1R correspond to the extracellular domains of MPXV proteins A35R and M1R, respectively. The human IgG1 Fc was inserted downstream of the mRNA constructs (eM1R, eA35R, and eM1-eA35) to enable Fc-mediated dimerization of the encoded proteins. SP represents the signal peptide of IL-6. The protein domains were connected by a flexible Glycine–Serine linker. (**b**) Detection of target protein expression via indirect IFA. Different mRNAs were transfected into 293T cells. Protein expressions were confirmed by IFA using a Flag antibody. Protein-expressing cells display green fluorescence. Scale bar: 100 μm. (**c**) Detection of target protein expression and dimer formation via Western blot using the Flag antibody. Cells transfected with mRNA were lysed and subjected to reduced and non-reduced conditions. Under reduced conditions, the bands appeared smaller, indicating the formation of a dimer. (**d**) Immunization schedule for mRNA vaccine candidates. Female BALB/c mice were intramuscularly injected with mRNA vaccines or a placebo, followed by a booster with the same dose three weeks later. Serum and splenocyte samples were collected at specified time points. (**e**,**f**) ELISA measurement of A35R- and M1R-specific IgG antibody titers. (**g**) PRNT measurement of neutralizing antibody levels against VACV-IMV in mouse serum. (**h**,**i**) ELISpot assessment of cellular immune responses induced by mRNA vaccine candidates. Splenocytes from BALB/c mice were harvested on day 14 after the final immunization and stimulated with peptide pools of A35R (**h**) and M1R (**i**) to measure the secretion of cytokines IFN-γ (**h**,**i**). Data are presented as mean ± SEM. Statistical significance was analyzed using the Mann–Whitney test or one-way ANOVA with multiple-comparison tests (ns, not significant, * *p* < 0.05, ** *p* < 0.01, *** *p* < 0.001, **** *p* < 0.0001).

**Figure 4 vaccines-13-00514-f004:**
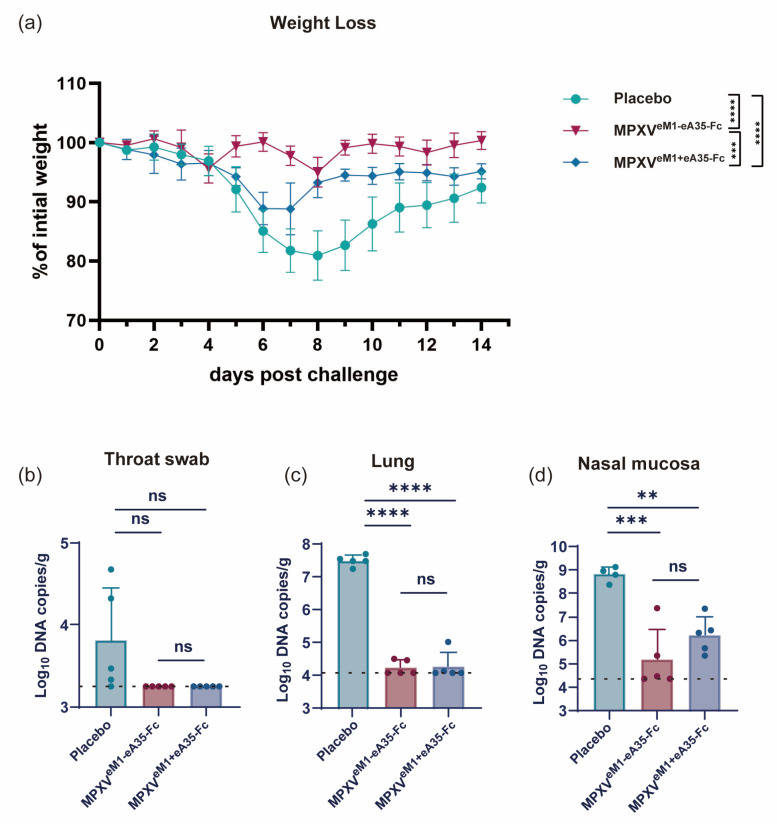
**mRNA vaccine candidates encoding dimeric antigens protect mice against VACV challenge:** (**a**) Mice immunized with the mRNA vaccine candidates or a placebo were intranasally inoculated with 10^5^ PFU of VACV, and changes in body weight were monitored over 14 dpi. (**b**–**d**) On day 7 after viral challenge, the mice were euthanized. Viral genome copies in throat swabs (**b**), lungs (**c**), and nasal mucosa (**d**) were detected by qPCR. Data are presented as mean ± SEM. Statistical significance was evaluated using two-way ANOVA with multiple-comparison test and one-way ANOVA (ns, not significant, ** *p* < 0.01, *** *p* < 0.001, and **** *p* < 0.0001).

## Data Availability

All data generated are presented in the manuscript.

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
