# Peer review of "A Single-Chain Mpox mRNA Vaccine Elicits Protective Immune Response in Mice"

_vaccines, 2025, doi:10.3390/vaccines13050514_

Round 1

Reviewer 1 Report

Comments and Suggestions for Authors

The manuscript prepared by Xu et al reports the development of a prototype of mpox mRNA vaccine based on lipid nanoparticles. Overall, the manuscript addresses a very relevant topic and contains new data suitable for publication in Vaccines. However, the manuscript needs to be revised before a decision can be made whether it can be accepted for publication.

1. Since mRNA vaccines require the use of delivery systems, it is necessary to discuss in the introduction which systems are described in the literature for this purpose. 
2. In the experimental part, the manufacturer of the lipids and the membrane filters used, including the membrane material, should be specified to exclude the question regarding whether lipid nanoparticles can be retained on the membranes during ultrafiltration. 
3. In the experimental part, the paragraph about vaccination and VACV challenge should be rewritten to make this paragraph more detailed and clear. 
4. Lines 181-182. The hydrodynamic particle diameter obtained by the DLS method should be rounded to whole values.
5. Figure 1b. Is the scale bar the same in all images? It would be convenient to put it on each photo separately. 
6. Discussion. It is necessary to discuss possible limitations of the developed mRNA vaccines. 

Reviewer 2 Report

Comments and Suggestions for Authors

The recent global outbreaks of the mpox virus (MPXV) underscore the urgent need for efficient next-generation vaccines. In this study, the authors aim to develop a single-chain mRNA vaccine encoding a fusion protein composed of two protective antigens: the ectodomains of M1R (eM1R) and A35R (eA35R) from MPXV. In mouse immunization and VACV challenge experiments, this fusion protein mRNA format elicited robust levels of neutralizing antibodies and antigen-specific cellular immune responses, outperforming the admixed formulation of the two individual antigens.

Additionally, the authors introduced a human IgG1 Fc domain to dimerize either the fusion vaccine or the monomeric antigen admixture, demonstrating that the Fc-dimerized fusion vaccine exhibits comparable or superior immunogenicity and protective efficacy compared to the admixed formats. The experimental design is relatively novel, and the results appear robust. This new mRNA vaccine platform shows promise for clinical application in MPXV vaccination and complements several recent publications in this field (e.g., PMID: 38366591, PMID: 40140411).

Major concerns:

  1. It is critical to include a longitudinal analysis of antibody responses following the booster immunization to demonstrate whether the fusion vaccine provides more sustained levels of virus-specific and neutralizing antibodies compared to the monomeric admixture formulation.
  2. Based on comparisons between Figure 1 and Figure 3, the antibody titers, VACV-IMV neutralization titers, and IFN-γ stimulation elicited by the dimerized format do not appear significantly higher than those from the monomeric fusion protein. To support the claim of superior immunogenicity and protection, a direct head-to-head immunization and challenge experiment comparing the monomeric and Fc-dimerized fusion vaccines should be included.
  3. The dose used in the VACV challenge experiments may need further optimization. In key readouts such as viral clearance from lung and nasal mucosa (Figure 2c–d and Figure 4c–d), no statistical differences are observed between the fusion vaccine and the monomeric admixture. Notably, other studies have used a higher challenge dose, such as 1.0 × 10⁷ PFU of MPXV (PMID: 40140411), compared to the 10⁵ PFU of VACV used in this manuscript.

Minor concerns:

  1. Figure 1a and 1b: The use of FLAG antibody to detect antigen expression in HEK293 cells should be clarified. Please indicate in Figure 1a where the FLAG tags are positioned in the antigen coding sequences. Additionally, describe whether a linker sequence was used in constructing the eM1R-eA35R fusion protein.
  2. Animal numbers: In all animal experiments, please specify the number of mice used per group.

Reviewer 3 Report

Comments and Suggestions for Authors

The authors evaluate the use of a single-chain mRNA vaccine to encode two antigens in mice to protect against mpox.

This is important to be able to simplify mRNA vaccine formulations with multiple antigens. The limitations of this study are the study is done only in mice and that it demonstrates the principle using only 2 antigens. It is likely that different antigens could behave differently. This should be discussed in the discussion. 

Line 171 for the eM1-eA35 construct was there any linker between the two proteins? Can this be clarified. Figure 1a shows some space between the proteins. Also what is SP in the constructs and where is the flag tag.

In the discussion.

Please discuss possible reasons for the lack of antibody responses to A35R in the eM1+eA35  vaccination. Is this due to the requirement both antigens being expressed in the same cell to allow for sufficient T cell responses to generate the antibody response to A25R? or some other mechanism?

Line 339 The superiority of the single-chain constructs is pushing the data a little. It might be better to state single-chain constructs are as effective and slightly more effective.

Round 2

Reviewer 2 Report

Comments and Suggestions for Authors

The authors have modified the text and addressed most of my concerns. I have no further comments.